# *Lignosus rhinocerotis* Cooke Ryvarden ameliorates airway inflammation, mucus hypersecretion and airway hyperresponsiveness in a murine model of asthma

**Malagobadan Johnathan[1], Siti Aminah Muhamad[1], Siew Hua Gan[2], Johnson Stanslas[3], Wan Ezumi Mohd Fuad[1], Faezahtul Arbaeyah Hussain[4], Wan Amir Nizam Wan Ahmad[1], Asma Abdullah Nurul[1] \***

**1** School of Health Sciences, Universiti Sains Malaysia, Kubang Kerian, Kelantan, Malaysia, **2** School of Pharmacy, Monash University Malaysia, Bandar Sunway, Selangor, Malaysia, **3** Pharmacotherapeutics Unit, Department of Medicine, Faculty of Medicine and Health Sciences, Universiti Putra Malaysia, Serdang, Selangor, Malaysia, **4** Department of Pathology, School of Medical Sciences, Universiti Sains Malaysia, Kubang Kerian, Kelantan, Malaysia

\* nurulasma@usm.my

## Abstract

*Lignosus rhinocerotis* Cooke. (*L. rhinocerotis*) is a medicinal mushroom traditionally used in the treatment of asthma and several other diseases by the indigenous communities in Malaysia. In this study, the effects of *L. rhinocerotis* on allergic airway inflammation and hyperresponsiveness were investigated. *L. rhinocerotis* extract (LRE) was prepared by hot water extraction using soxhlet. Airway hyperresponsiveness (AHR) study was performed in house dust mite (HDM)-induced asthma in Balb/c mice while airway inflammation study was performed in ovalbumin (OVA)-induced asthma in Sprague-Dawley rats. Treatment with different doses of LRE (125, 250 and 500 mg/kg) significantly inhibited AHR in HDM-induced mice. Treatment with LRE also significantly decreased the elevated IgE in serum, Th2 cytokines in bronchoalveolar lavage fluid and ameliorated OVA-induced histological changes in rats by attenuating leukocyte infiltration, mucus hypersecretion and goblet cell hyperplasia in the lungs. LRE also significantly reduced the number of eosinophils and neutrophils in BALF. Interestingly, a significant reduction of the FOXP3$^+$ regulatory T lymphocytes was observed following OVA induction, but the cells were significantly elevated with LRE treatment. Subsequent analyses on gene expression revealed regulation of several important genes i.e. *IL17A*, *ADAM33*, *CCL5*, *IL4*, *CCR3*, *CCR8*, *PMCH*, *CCL22*, *IFNG*, *CCL17*, *CCR4*, *PRG2*, *FCER1A*, *CLCA1*, *CHIA* and *Cma1* which were up-regulated following OVA induction but down-regulated following treatment with LRE. In conclusion, LRE alleviates allergy airway inflammation and hyperresponsiveness, thus suggesting its therapeutic potential as a new armamentarium against allergic asthma.

**Data Availability Statement:** All relevant data are within the manuscript and its Supporting Information files.

**Funding:** This work was supported by the Universiti Sains Malaysia Research University Grants (www.usm.my) (Grant no: 1001/PPSG/813065 and 1001/PPSK/812180). The grants were awarded to Nurul AA. The funders had no role in study design, data collection and analysis, decision to publish, or preparation of the manuscript.

**Competing interests:** The authors declare that they have no conflict of interest.

## Introduction

Asthma is a chronic inflammatory respiratory disease, as characterized by an airflow obstruction, airway inflammation and hyperresponsiveness [1]. During allergen stimulation, dendritic cells induce the differentiation of T cells into Th2 cells [2]. Th2 cells secrete cytokines i.e. interleukin (IL)-4, IL-5, IL-9 and IL-13 which facilitate recruitment and activation of eosinophils in the airway [3]. IL-5 and IL-9 are critical for promoting tissue eosinophilia and mast cell hyperplasia, whereas IL-13 stimulates mucus production by goblet cells and airway hyperresponsiveness (AHR) [4]. Meanwhile, Th2 cells will induce the production of IgE by the B cells via IL-4 stimulation. In the presence of airway inflammation, the cellular components especially the eosinophils and mast cells will be prompted to release a number of different mediators with the capacity to cause AHR [5]. The inflammatory mediators such as IL-13, histamine, major basic proteins (MBP) and leukotrienes are known to cause AHR which lead to bronchoconstriction, and finally remodelling of the lung.

The $CD4^+CD25^+Foxp3^+$ regulatory T cells (Treg) are crucial in regulating Th2-induced allergic responses [6]. Kearley et al. demonstrated that therapeutic transfer of $CD4^+CD25^+$Treg can curb allergen-induced inflammation and preclude airway remodelling [7]. Therefore, targeting T cell, specifically on $FOXp3^+$Tregs offers a good promise to asthma treatment

Changes in gene expression on airways structural cells are purported to be important in the progress of asthma and AHR. Some of the changes include alterations to inflammatory, smooth muscle contractility and epithelial barrier integrity genes. Some of the genes for example IL-4 receptor and ADAM metallopeptidase domain 33 (ADAM33) have been linked to asthma [8]. These genes have steadily shown similar link to asthma phenotypes in diversified populations and are therefore purported to contribute mainly to the pathogenesis of asthma [9].

Current treatments for asthma are categorised into two; long-term control medications such as inhaled corticosteroids and leukotriene modifiers that control inflammation in the airway, while quick-relief medications such as β2 adrenergic agonists and anti-cholinergic agents that cause airway smooth muscle relaxation. Nevertheless, they are not without various side effects [10], necessitating a safer alternative for asthma management may confer a promising step.

*Lignosus rhinocerotis* (Cooke) Ryvarden (*L. rhinocerotis*) is a type of medicinal mushroom which belongs to the Polyporaceae family and is primarily found in the tropical Asian forests [11]. Scientifically, studies have confirmed its anti-proliferative, antioxidant, antimicrobial, anti-inflammatory and immunomodulatory effects [12], indicating its vast unexplored potentials. Furthermore, this mushroom has been widely used as a traditional medicine to treat cough and asthma. It was previously reported that the Temuan tribe in Malaysia used the mushroom by boiling its sclerotium with other herbs to reduce cough and asthma symptoms [13]. Previous studies reported that oral and intranasal administrations of *L. rhinocerotis* extract (LRE) had attenuated asthmatic parameters by reducing Th2-type cytokines level, IgE and leukocyte infiltration in the lungs [14, 15].

Sensitisation methods utilising ovalbumin (OVA) and house dust mite (HDM) are known to enhance manifestations of asthmatic features. OVA- and HDM-induced models are considered the appropriate methods for experimental allergic asthma. These models have similar clinical symptoms to human asthma, which are characterized by airway inflammation, thickening of bronchial wall, mucus hypersecretion and increased infiltration of inflammatory cells into the lung [16]. However, there has been limited success with OVA-induced model and only moderate pulmonary inflammation and mild AHR have been observed [17]. HDM has become more commonly used in mouse models to induce AHR because of its immunogenic

properties, so the use of an adjuvant is not required [18]. In addition, inhaled delivery of HDM has been more successful in inducing AHR, possibly because of the intrinsic enzymatic activity of this allergen. In this study, LRE was administrated orally into OVA- and HDM-induced asthmatic models. The potential effect of LRE oral administration in ameliorating airway inflammation, hyperresponsiveness and mucus hypersecretion as well as regulating cellular components and gene expression in the airway of allergic asthma model is further investigated in this study.

## Materials and methods

### *L. rhinocerotis* extraction

The sclerotia of *L. rhinocerotis* cultivar was obtained in dried powdered formulation from a local company named Ligno Biotech Sdn. Bhd. Malaysia. For preparation of extract, *L. rhinocerotis* sclerotium powder (150 g) was first soaked in purified distilled water (600 ml) and subjected to 24-hour hot water extraction using a soxhlet extractor. The process was followed by a drying step via a rotary evaporator (Ilshin BioBase, Gyeonggi-do, South Korea) followed by freeze-drying to produce lyophilized powder. Generally, 50 g of sclerotial powder produced 5 g of LRE (i.e a 10% yield).

### House Dust Mite (HDM)–induced mouse model of asthma

Ethical approval was obtained from the Institutional Animal Care and Use Committee (IACUC) of the Universiti Putra Malaysia (UPM/IACUC/AUP-R056/2017). In this part of the study, 6 to 8 weeks old female Balb/c mice were used. The animals were randomised into six groups (n = 6/groups): (1) normal control (2) HDM, sensitised and challenged with HDM (3) LRE125, sensitised and challenged with HDM; treated with oral LRE (125 mg/kg) (4) LRE250, sensitised and challenged with HDM; treated with oral LRE (250 mg/kg) (5) LRE500, sensitised and challenged with HDM; treated with oral LRE (500 mg/kg) (6) Dex, sensitised and challenged with HDM; treated with i.p dexamethasone (3 mg/kg). Sensitisation was done by intranasal (i.n) administration of 100 μg HDM (Stallergenes Greer, USA) prepared in phosphate-buffered saline (PBS) in which 10 μL was administered to each nostril (20 μL/mouse) at baseline (day 0). Subsequently, daily intranasal challenges (10 μg HDM in 20 μL PBS) were incorporated on days 7, 8, 9, 10 and 11 based on Hammad et al. in which treatments were instituted one hour before the challenge [19].

Finally, the mice were anesthetized on day 14, by means of ketamine (100 mg/kg) and xylazine (10 mg/kg) injections. The procedure ended with a tracheotomy.

### Measurement of airway hyperresponsiveness

On Day 14, measurement of AHR was conducted. Tracheotomy was conducted on the mice following anaesthesia (100 mg/kg ketamine and 10 mg/kg xylazine). Briefly, the trachea was cannulated with a blunt end (20 G) to join to the nebulizer in order to facilitate administration of 1x PBS and methacholine. Each animal was kept in a whole-body plethysmograph chamber and was mechanically ventilated (200 μL/breath tidal volume) at 150/min. Airflow was monitored using transducers, while pressure changes was detected via a recorder. The findings were analysed using a FinePointe™ RC System (Buxco Research Systems, New Brighton, USA). Airway resistance (RI) was monitored in response to various methacholine concentrations (1, 2, 4, 8, 16 and 32 mg/mL). Subsequently, PBS and methacholine were continuously added to the nebulizer at 4 min intervals. The findings were expressed as percentage of the respective basal values against PBS.

## Ovalbumin (OVA)-induced rat model of asthma

The study was conducted at Universiti Sains Malaysia (USM) following ethical approval from USM Animal Ethics Committee (AEC/2012/(78)(397). Male Sprague Dawley rats age 6 to 8 weeks old were utilised. The animals were randomised into six groups (n = 5 /group): 1) normal group 2) OVA, sensitised and challenged with 1% OVA 3) LRE125, sensitised and challenged with OVA; treated with oral LRE (125 mg/kg) (4) LRE250, sensitised and challenged with OVA; treated with oral LRE (250 mg/kg) (5) LRE500, sensitised and challenged with OVA; treated with oral LRE (500 mg/kg) (4) Dex, LRE125, sensitised and challenged with OVA; treated with dexamethasone (3 mg/kg). LRE dose of 500 mg/kg was selected for subsequent experiments based on the data generated from dose-response study.

Sensitisation was conducted on days 1 and 14 by intraperitoneal (i.p.) injection of OVA (10 mg/ml) and aluminium hydroxide (alum) (100 mg/ml) each in phosphate buffer saline (PBS) [16]. Subsequently, *Bordetella pertussis* (50 ng/ml) was included in the solution. The rats were challenged with 1% OVA aerosol on day 23. The duration of challenge was for 20 min/day by an ultrasonic nebulizer coupled with oral administration of LRE (500 mg/kg) via oral gavaging (Mabist mist, Illinois, USA) for seven days. The rats were sacrificed on the following day, followed by collection of lungs, serum and BALF for downstream analyses.

## Measurement of total IgE in serum

The total IgE was measured with a mouse specific ELISA kit (Abnova, Heidelberg, Germany) according to the manufacturer's instructions. IgE level in each sample was measured using optical density at 450 nm followed by calculation based on a standard curve generated using recombinant IgE.

## Measurement of Th2 cytokines in BALF

ELISA was performed according to the manufacturer's instructions. IL-4, IL-5 and IL-13 in BALF were measured using specific rat IL-4, IL-5 and IL-13 ELISA kits (CusaBio, Hubei, China). Absorbance was measured at 450 nm. Cytokine concentrations were calculated from standard curves that were generated using respective recombinant interleukins.

## Histological analysis of lung

Lungs were fixed in formalin (10%), dehydrated in various concentrations of ethanol, embedded in paraffin and cut into 4 μm sections. The tissue sections were deparaffinized and rehydrated using xylene and graded alcohol respectively, and stained with hematoxylin and eosin (H&E) solution or with periodic acid Schiff (PAS) stain. Lung inflammation score was assessed as the following scale: 0, no inflammation; 1, mild inflammation; 2, moderate inflammation; 3, marked inflammation; and 4, severe inflammation [20]. Goblet-cell hyperplasia in airway epithelium was calculated based on a five-point system where "0": no goblet cells, "1": <25% of epithelium, "2": 25–50% of epithelium, "3": 51–75% of epithelium and "4" >76% of epithelium [21]. The mean scores for mucus production were similarly calculated by means of a Mirax Image viewer (Carl Zeiss, Germany).

## Cell subsets analysis by flow cytometry

Bronchoalveolar lavage pellet containing approximately $1 \times 10^6$ total cells were stained with CD4[+] FITC, CD25[+] PE and Foxp3[+] APC antibodies to detect T-reg cells. To detect eosinophils, the cells were stained with a combination of CD3[+] PE (1 μg/ml)/RP-1[+] PE (1 μg/ml)/

HIS48[+] FITC (1 µg/ml) as described previously [17], the cells were resuspended in PBS and were subjected to a BD Accuri™ C6 flow cytometer.

## RNA isolation and gene array analysis

RNA was isolated from the lung tissues. Subsequently, the tissues were stabilized at -80˚C in RNAlater solution (Qiagen, USA). RNeasy® Mini kit was used for RNA extraction as recommended by the manufacturer (Qiagen, USA). The extracted RNA from each sample (5 µg) was reverse transcribed to cDNA in a final volume of 20 µl using RT[2] First strand kit.

In the study, the RT[2] Profiler polymerase chain reaction (PCR) Array (Qiagen, USA) for allergy and asthma pathway-related genes was utilised. It estimates the expression of key genes (n = 84) that regulate allergy mechanism. The value for threshold cycle (Ct) of each gene was estimated as a relative value following normalization against the housekeeping genes (β-actin (*Actb*), beta-2 microglobulin (*B2m*), ribosomal protein large P1 (*Rplp1*), lactate dehydrogenase A (*Ldha*) and hypoxanthine phosphoribosyltransferase 1 (*Hprt1*). Data analysis was conducted based on the manufacturer's web portal for PCR array data analysis. The relative gene expression level was estimated using the $2^{-\Delta\Delta Ct}$ method with the value expressed as fold regulation as opposed to the control.

## Statistical analysis

Data were expressed as mean ± standard deviation (SD). Statistical significance (p<0.05) was determined by a one-way ANOVA test followed by Scheffe's post hoc correction using Statistical Programme for Social Science (SPSS) version 20.0 (New York, USA). For inflammation and mucus scoring, statistical analyses were performed using Kruskal Wallis tests with confidence interval adjustment by Bonferroni correction. The data from PCR array was analysed by the web-based software (SABiosciences) using a 2–ΔΔCT method. The expression level for genes of interest was normalized by the expression of housekeeping genes.

## Results

### Effects of LRE on methacoline-induced AHR

Following investigation of airway responsiveness to methacoline in each group, HDM-challenged mice demonstrated elevated AHR to methacoline (p<0.05) as confirmed by the increased in RI values when compared to normal (Fig 1). Nevertheless, pre-treatment with dexamethasone and LRE (125, 250 and 500 mg/kg) significantly attenuated RI levels (where the concentrations were maintained at lower than 8 mg/ml) as compared to untreated HDM-group (higher than 8 mg/ml in all of the investigated methacoline concentrations).

### Effects of LRE on IgE production in serum

As shown in Fig 2, the OVA-induced asthmatic rats (OVA group) presented significantly increased IgE levels compared to normal group. Significant decreased in the mean serum of IgE level was observed after treatments with LRE at 250 and 500 mg/kg (p < 0.05) when compared with OVA group. However, no significant decrease could be observed from the rats treated with LRE 125 mg/kg (Fig 2).

### Effects of LRE on Th2 cytokines in BALF

As shown in Fig 3, the levels of IL-4, IL-5 and IL-13 were significantly increased in BALF of OVA group. Treatment with LRE 500 mg/kg resulted in significantly decreased of IL-4, IL-5 and IL-13 levels when compared to OVA group. In contrast, treatment with LRE 250 mg/kg

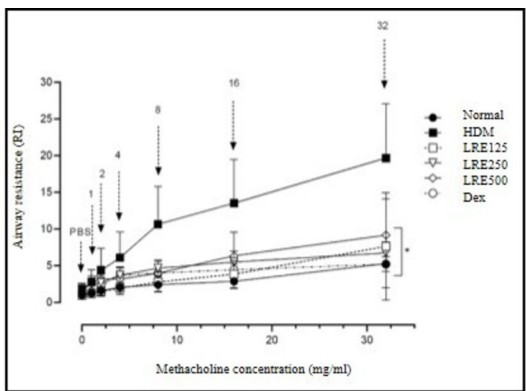

**Fig 1. Suppression of airway hyper-responsiveness after administration of LRE in HDM-induced mouse model of asthma.** (A) Airway resistance was measured by whole body plethysmography. Data are expressed as mean ± SD (n = 5). *p<0.05 is significantly different from HDM group.

significantly decreased only IL-5 level, while LRE 125 mg/kg significantly decreased IL-4 and IL-5 levels.

## Effects of LRE on histopathological change in lung tissue

Histopathological analysis by H&E staining demonstrated that lung tissue exposed with OVA had a marked increase in the leukocyte infiltration into the lung tissues. Treatment with LRE 250 or LRE 500 mg/kg significantly decreased infiltration of leukocytes in the peribronchial region and perivascular connective tissue of the airway compared with that observed in the OVA group (Fig 4). No significant attenuation was observed by the LRE 125 mg/kg treatment group. Based on the dose-response study, LRE at the 500 mg/kg concentration was selected for the downstream experiments.

## Effects of LRE on goblet cell hyperplasia

The number of PAS positive cells increased markedly in the lungs section of the OVA group when compared to the normal (Fig 5) indicating higher mucus production in the lungs. For the group which received LRE, the number of PAS positive cells in the lungs section were significantly reduced when compared to that in the OVA group. Similarly, mucus production was also decreased in animals treated with dexamethasone. Quantitative scoring of mucus

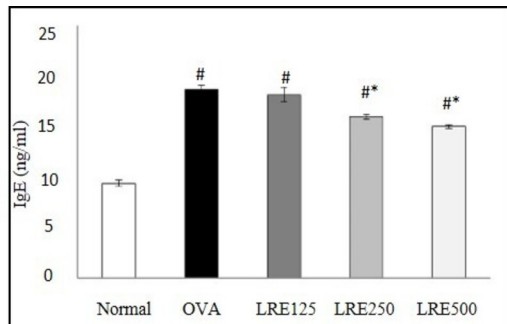

**Fig 2. Effects of different doses of LRE (125, 250, 500 mg/kg) on serum IgE levels.** Data are expressed as means ± SD (n = 5/group). # p < 0.05 indicates significance difference from the normal group. * p < 0.05 indicates significant difference from OVA control group.

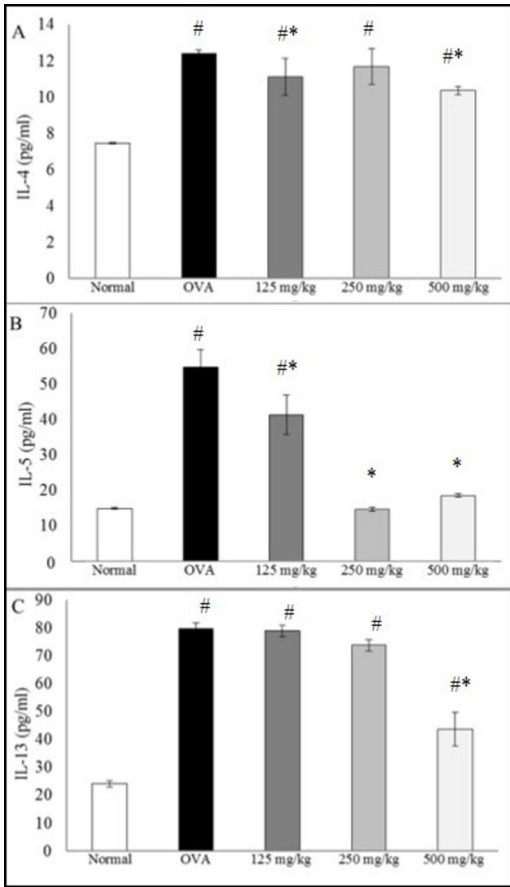

**Fig 3. Effects of different doses of LRE (125, 250, 500 mg/kg) on Th2 cytokines level in BALF.** Data are expressed as means ± SD (n = 5 per group). # p < 0.05 indicates significance difference from the normal group. * p < 0.05 indicates significant difference from OVA control group.

production suggested that both LRE and dexamethasone significantly (p < 0.01) reduced mucus production, indicating LRE as a potential treatment in ameliorating the severity of allergic asthma.

## LRE modulates leukocytes compositions

The populations of lymphocyte, neutrophil and eosinophil in BALF cells were evaluated by flow cytometry analysis. Fig 6 shows significant elevation of CD4⁺ cells population in the BALF following OVA induction. Treatment with LRE significantly reduced the percentage of CD4⁺ cells population, similar to the dexamethasone group, but the CD4⁺ cells population was still higher than the normal group. Lymphocyte populations were further evaluated by analysing CD4⁺CD25⁺Foxp3⁺ Treg cells. There was a significant reduction of the Foxp3⁺ Treg population following OVA induction when compared to the normal group (Fig 7). Interestingly, following LRE treatment, Treg population was significantly elevated similar as that seen in the dexamethasone group, demonstrating the potential effect of LRE in ameliorating Foxp3⁺ Treg population. Based on the flow cytometric analysis of CD3⁺T-cells, HIS48⁺ and RP-1⁺ cell populations in BALF, it was noteworthy that eosinophils population was highly increased in the OVA group as compared to the normal group (Fig 8C). However, treatment with LRE inhibited eosinophil levels as that also seen with dexamethasone. Additionally, there was significant

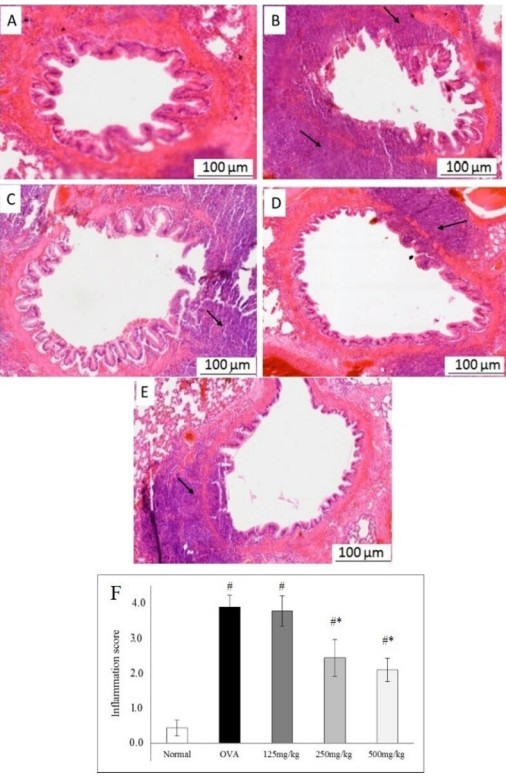

**Fig 4. Effects of different doses of LRE (125, 250, 500 mg/kg) on airway inflammation in the peribronchiole region and perivascular connective tissue in OVA-induced allergic asthma model (H&E staining, original magnification 50×).** Representative photomicrographs showing (A) Normal; (B) OVA control group; (C) LRE 125 mg/kg; (D) LRE 250 mg/kg; (F) LRE 500 mg/kg (F) Graphs represent inflammation score. Black arrows indicate the presence of infiltration with eosinophils or leukocytes surrounding the bronchiole.

reductions in neutrophils population in all treatment groups when compared to the OVA group (Fig 8D).

## LRE-regulated allergic asthma gene expression

A scatter plot of gene expression from the lung tissues of OVA- and LRE-treated groups was plotted (Fig 9). Based on the figure, a total of 16 genes were significantly ($p < 0.05$) up-regulated following sensitization with OVA (Table 1). In fact, the genes (*Il17a, Adam33, Chia, IL4, Ccr3, Ccl5, Ccl17, Pmch, Clca1, Cma1, Ccr8, Fcer1a, Ifng, Ccr4, Ccl22* and *Prg2)* were differentially regulated by more than 1.5 fold and were expressed at a relatively high mean threshold cycle (amplified at cycle <30). Interestingly, the genes that were up-regulated by OVA-sensitization were significantly further down-regulated upon administration of LRE (Table 2) although some genes were not affected. On the other hand, *IL-5, Arg1, Foxp3 and Ear11* were down-regulated with LRE treatment. In contrast, *GATA3* showed an up-regulation pattern following LRE treatment.

## Discussions

Allergic asthma is a multifactorial disease involving numerous clinical symptoms including airway inflammation, manifested as airway hyperresponsiveness, elevated IgE level and Th2 cytokines, goblet cell hyperplasia, excessive mucus secretion and eosinophil infiltration.

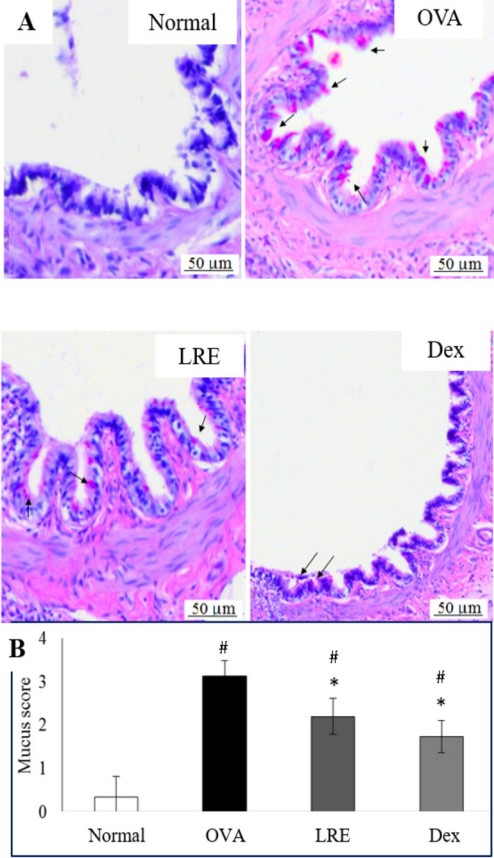

**Fig 5. Attenuation of mucus production and goblet cell hyperplasia following administration of LRE in OVA-induced rat model of asthma.** (A) Representative photomicrographs showing periodic acid–Schiff (PAS) staining (magnification ×40) staining for measuring mucus production in the airways. The black arrows indicate the presence of stained goblet cells. (B) Quantitative analysis of scoring on mucus production was done by scoring with a subjective scale of 0–4. Data was expressed as mean ± SD (n = 5 per group). # p < 0.05 indicates significance difference from the normal group. * p < 0.05 indicates significant difference from OVA control group.

Previous studies reported the anti-asthmatic effects of LRE to be occurring via attenuation of inflammatory cell infiltration in the lungs tissue, reduction in Th2 cytokines (IL-4, IL-5 and IL-13) in the BALF as well as reduction in serum IgE levels [14, 15]. In this study, oral administration of LRE at different doses (125, 250 and 500 mg/kg) showed varying responses. Among the doses tested in this study, LRE at 500 mg/kg significantly attenuated the level of IgE in serum, Th2 cytokines IL-4, IL-5 and IL-13 in BALF as well as leukocyte infiltrations in the lung tissues, in contrast to the other two lower doses, 125 and 20 mg/kg which were not consistently attenuated the responses. Based on the dose-response study, LRE 500 mg/kg was chosen for the subsequent studies.

Airway inflammation is the basis of AHR, and the process involves both cellular and non-cellular aspects of airway inflammation [22]. It has been well documented that IL-4 and IL-5 play critical roles in airway inflammation by mobilising and activating eosinophils, which lead to the release of pro-inflammatory mediators such as major basic protein and cysteinyl leuko-trienes, which have the capacity to cause AHR [23]. In the present study, short-term inhalation of HDM resulted in a marked increase in airway resistance (RI) towards increasing doses of metacholine in comparison to normal group. It was in agreement with Johnson *et al.*, [24], which indicated that HDM exposure resulted in increasing airway resistance but reduced

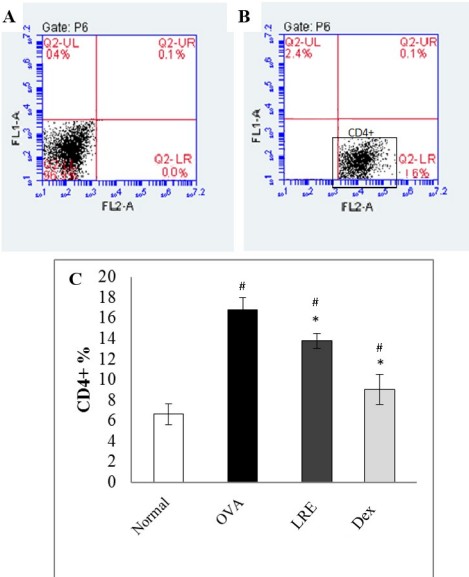

**Fig 6. The effects of LRE on the CD4+ population of BALF cells in OVA-induced rat model of asthma.**
Representative dot plot showing (A) unstained (B) CD4+ cells. Data was expressed as mean ± SD (n = 5 per group). #
p < 0.05 indicates significance difference from the normal group. *p < 0.05 indicates significant difference from OVA
control group.

dynamic compliance (Cdyn), indicated that airway inflammation and remodelling contribute
to AHR. In this study, treatment with LRE and dexamethasone has led to a dramatic decrease
in the elevated AHR towards increasing concentration of metacholine in asthmatic mice in the
range of 8–32 mg/ml. The alleviation of AHR in HDM-induced mice may be associated with
the reduction of inflammatory cells and mediators of allergic asthma measured in the study.

Although the primary roles of mucus in the respiratory tract is to protect lower airways
from dehydration and impairment, it is no doubt that hyperplastic goblet cells producing
excessive mucus may exacerbate asthmatic condition. It is also a known fact that upon exacer-
bation of asthma, the resulting response which induces IgE switching leads to mucus hyperse-
cretion from the goblet cells [25]. In our study, the number of PAS positive cells was highly
alleviated in the lung section of animals which received LRE as compared to OVA. Similar suc-
cessful anti-asthmatic effects have been reported in many plant extracts such as *Kochia sco-
paria* fruit extract and *Cassia occidentalis* which attenuated the goblet cells number [25, 26]

Airway inflammation is demonstrated by increased levels of a variety of immune cells and
inflammatory mediators including eosinophils and lymphocyte [3]. Eosinophilic infiltration in
the lungs tissue is the hallmark of airway inflammation in asthma. In this study, we demon-
strated reduced number of eosinophils and neutrophils (p<0.05) in the BALF following LRE
and dexamethasone administrations. Similar findings on attenuation of the eosinophil infiltra-
tion in the peribronchial region of the lungs were reported previously [14, 15]. Previously,
Green et al. reported suppression of eosinophil and T-lymphocyte inflammation by corticoste-
roids without preventing neutrophil accumulation indicating that the two markers (eosinophil
and T-lymphocyte) are important in resistant development [27]. Meanwhile, An et al. reported
no significant reduction in alveolar neutrophils, eosinophils and lymphocytes numbers with
dexamethasone or clarithromycin administration in neutrophilic asthma mouse model, but
with combined treatments, the numbers of neutrophils decreased (p<0.05) [28], indicating

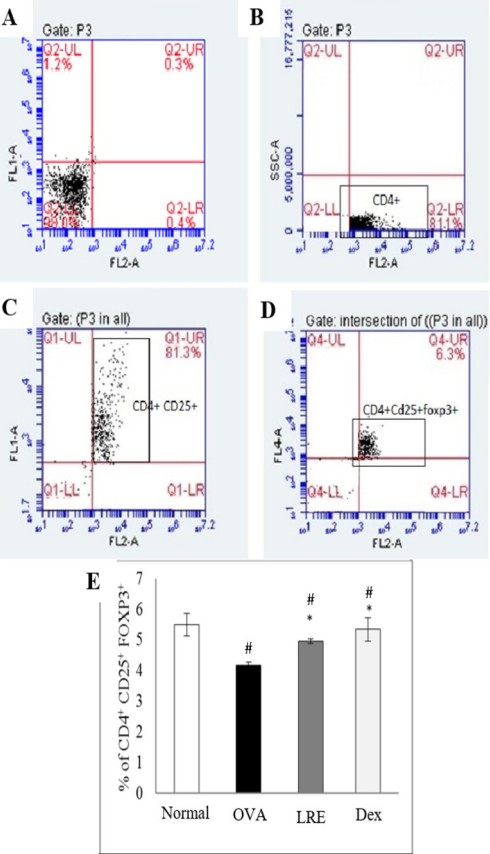

**Fig 7. The effects of LRE on the CD4+CD25+Foxp3+ Treg population in BALF of OVA-induced rat model of asthma.** Representative dot plot showing (A) unstained, (B) CD4+ cells, (C) CD4+CD25+ cells and (D) CD4+CD25 +Foxp3+ Treg cells. Data was expressed as mean ± SD (n = 5 per group). # p < 0.05 indicates significance difference from the normal group. * p < 0.05 indicates significant difference from OVA control group.

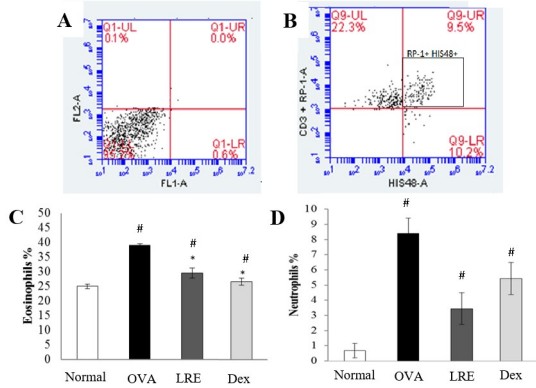

**Fig 8. The effects of LRE on eosinophils and neutrophils population in BALF of OVA-induced rat model of asthma.** Representative dot plot showing (A) unstained and (B) RP-1+ HIS48+ positive cells. BALF was stained with specific antibodies for CD3 FITC (T-cells), HIS48 FITC stained to detect (C) eosinophils population and RP-1, PE stained to detect (D) neutrophils population. Data was expressed as mean ± SD (n = 5 per group). # p < 0.05 indicates significance difference from the normal group. * p < 0.05 indicates significance different from OVA control group.

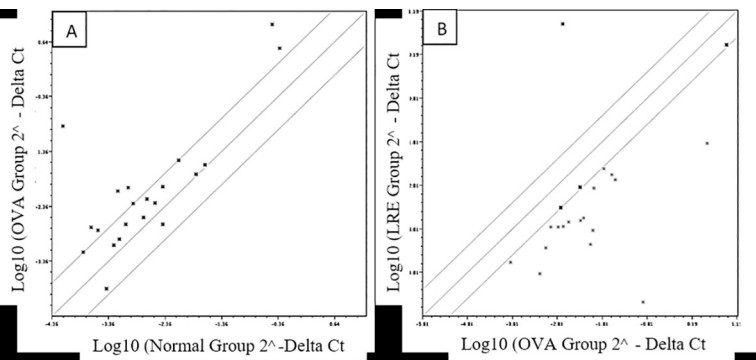

**Fig 9.** Scatter plots of asthma genes expression of the lungs tissue in OVA-induced rat model of asthma following (A) OVA sensitization and (B) OVA sensitization + LRE. Fold-change values of less than one indicates a down-regulation and that the fold-regulation is the negative inverse of the fold-change.

that some agents may act synergistically and may need to be combined or to be used as an adjuvant.

Flow cytometry analysis in BALF indicated a significant increase in CD4⁺ population which signifies inflammation upon OVA sensitization when compared to the normal group. However, LRE reduced the percentage of CD4⁺ T cells significantly. Similar finding was also reported by Liu et al., where the effects of polysaccharides *Antrodia camphorate* mushroom strongly inhibited both antigen-specific and non-antigen-specific T-cell immune responses [29]. In general, CD4⁺ contributes to enhancement and proliferation of Th2 cells, which can be a useful step in ameliorating asthma.

Significant roles of CD4⁺CD25⁺Foxp3⁺ Treg cells have previously been identified in asthma regulation. We demonstrated reduction of CD4⁺CD25⁺Foxp3⁺Treg cells following OVA induction; but increased CD25⁺ Foxp3 population in LRE-treated group which was similarly seen as with dexamethasone. In a previous study, treatment with erythromycin caused reduction in inflammatory infiltrates in the BALF and attenuated lung damages although it

**Table 1. Up-regulation of genes following OVA induction in rats.**

| Gene symbol/ID | Fold change | Gene description |
| --- | --- | --- |
| **Up-regulated genes** | | |
| *IL17A* | 414.07 | Interleukin 17A |
| *ADAM33* | 61.92 | ADAM metallopeptidase domain 33 |
| *CHIA* | 26.37 | Chitinase, acidic |
| *IL4* | 9.55 | Interleukin 4 |
| *CCR3* | 8.15 | Chemokine (C-C motif) receptor 3 |
| *CCL5* | 7.04 | Chemokine (C-C motif) ligand 5 |
| *CCL17* | 3.96 | Chemokine (C-C motif) ligand 17 |
| *PMCH* | 3.19 | Pro-melanin-concentrating hormone |
| *CLCA1* | 3.19 | Chloride channel accessory 1 |
| *Cma1* | 3.13 | Chymase 1, mast cell |
| *CCR8* | 2.62 | Chemokine (C-C motif) receptor 8 |
| *FCER1A* | 2.51 | Fc fragment of IgE, high affinity I, alpha receptor |
| *IFNG* | 2.36 | Interferon gamma |
| *CCR4* | 2.29 | Chemokine (C-C motif) receptor 4 |
| *CCL22* | 2.16 | Chemokine (C-C motif) ligand 22 |
| *PRG2* | 2.09 | Proteoglycan 2, bone marrow |

**Table 2. Up and down-regulation of genes following LRE administration in OVA-induced asthma in rats.**

| Gene symbol/ID | Fold change | Gene description |
|---|---|---|
| **Up-regulated genes** | | |
| *GATA3* | 3801.69 | GATA binding protein 3 |
| **Down-regulated genes** | | |
| *IL17A* | -39241.53 | Interleukin 17A |
| *ADAM33* | -662.13 | ADAM metallopeptidase domain 33 |
| *CCL5* | -231.78 | Chemokine (C-C motif) ligand 5 |
| *IL4* | -122.46 | Interleukin 4 |
| *CCR3* | -67.77 | Chemokine (C-C motif) receptor 3 |
| *CCR8* | -43.77 | Chemokine (C-C motif) receptor 8 |
| *PMCH* | -21.27 | Pro-melanin-concentrating hormone |
| *CCL22* | -21.11 | Chemokine (C-C motif) ligand 22 |
| *IFNG* | -15.16 | Interferon gamma |
| *CCL17* | -14.48 | Chemokine (C-C motif) ligand 17 |
| *FOXP3* | -12.48 | Forkhead box P3 |
| *CCR4* | -11.65 | Chemokine (C-C motif) receptor 4 |
| *PRG2* | -9.43 | Proteoglycan 2, bone marrow |
| *Arg1* | -9.31 | Arginase |
| *FCER1A* | -7.53 | Fc fragment IgE, high affinity I, receptor alpha polypeptide |
| *IL5* | -5.24 | Interleukin 5 |
| *EAR11* | -4.57 | Eosinophil-associated, ribonuclease A, member 11 |
| *CLCA1* | -3.88 | Chloride channel accessory 1 |
| *Chia* | -3.52 | Chitinase, acidic |
| *Cma1* | -3.50 | Chymase 1, mast cell |

increased CD4$^+$Foxp3$^+$ Tregs number in the lungs [30]. A recent study by Kim et al. demonstrated elevated percentage of CD4$^+$CD25$^+$ cells in the BALF of OVA-induced mice which was reduced following treatment with *Anthriscus sylvestris* root extract [31]. In fact, the detectable changes in the range of Tregs cells in our study is in accordance with the ranges reported in previous studies [30, 32].

PCR array analysis demonstrated significant upregulation of a total of 16 genes, related to asthma namely *IL-17A*, *ADAM33*, *CHIA*, *IL4*, *CCR3*, *CCL5*, *CCL17*, *PMCH*, *CLCA1*, *Cma1*, *CCR8*, *FCER1A*, *IFNG*, *CCR4*, *CCL22* AND *PRG2* following OVA sensitization but downregulated following LRE treatment (Table 2). These genes are among the key genes central to allergic and asthmatic responses and they are important for the activation and cellular responses of Th2 cells, mast cells, eosinophils, NK cells and alternatively-activated macrophages [33, 34]. Among the genes, IL17A and ADAM33 have been implicated in the severity of asthma. Several studies have shown elevated IL17A mRNA expression in severe asthma [31, 35, 36], some of which have reported IL17A signalling involvement in the mobilisation of neutrophils and smooth muscle cells in the airways [37–39]. ADAM33 expression was demonstrated to increase as asthma severity increases [40, 41], which responsible to promote airway smooth muscle thickening thus leads to airway remodelling [42]. Our future studies may focus on the bioactive compositions in LRE and their molecular mechanisms underlying the anti-asthmatic effects in the airway inflammation model.

## Conclusion

Taken together, the findings in this study demonstrated that *L. rhinocerotis* extract significantly attenuated IgE, Th2 cytokines, leukocyte infiltration and mucus producing goblet cells

in the lung epithelium, alleviated airway hyperresponsiveness and down-regulated selected genes which are important in the regulation of allergy asthma. In addition, the population of FOXP3$^+$ Treg cells was elevated while that for eosinophils and neutrophils were reduced in BALF. This study reports the anti-asthmatic effects of *L. rhinocerotis* in airway inflammation of murine model, indicating a new potential alternative for the management of asthma.

## Supporting information

**S1 Checklist.**
(PDF)

**S1 File.**
(DOCX)

## Acknowledgments

The authors would like to thank LignoBiotech™ Sdn. Bhd and the staff from Animal Research and Service Centre Universiti Sains Malaysia.

## Author Contributions

**Conceptualization:** Asma Abdullah Nurul.

**Data curation:** Wan Amir Nizam Wan Ahmad.

**Formal analysis:** Malagobadan Johnathan.

**Funding acquisition:** Asma Abdullah Nurul.

**Investigation:** Malagobadan Johnathan, Siti Aminah Muhamad.

**Methodology:** Siew Hua Gan, Johnson Stanslas, Faezahtul Arbaeyah Hussain, Asma Abdullah Nurul.

**Project administration:** Asma Abdullah Nurul.

**Supervision:** Siew Hua Gan, Wan Ezumi Mohd Fuad, Asma Abdullah Nurul.

**Validation:** Faezahtul Arbaeyah Hussain, Asma Abdullah Nurul.

**Writing – original draft:** Malagobadan Johnathan.

**Writing – review & editing:** Wan Ezumi Mohd Fuad, Asma Abdullah Nurul.

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
