## [Decision Letter · Decision Letter 0]

8 Jan 2021

PONE-D-20-35276

Lignosus rhinocerotis Cooke Ryvarden ameliorates airway inflammation, mucus hypersecretion and airway hyper-responsiveness in a murine model of asthma

PLOS ONE

Dear Dr. Nurul,

Thank you for submitting your manuscript to PLOS ONE. After careful consideration, we feel that it has merit but does not fully meet PLOS ONE’s publication criteria as it currently stands. Therefore, we invite you to submit a revised version of the manuscript that addresses the points raised during the review process.

While the reviewer #1 was less critical, the comments from the reviewer #2 were very important. The authors need to effectively respond to their comments in their revision.

We look forward to receiving your revised manuscript.

Kind regards,

Yu Ru Kou, PhD

Academic Editor

PLOS ONE

Journal Requirements:

2. As part of your revision, please complete and submit a copy of the Full ARRIVE 2.0 Guidelines checklist, a document that aims to improve experimental reporting and reproducibility of animal studies for purposes of post-publication data analysis and reproducibility: https://arriveguidelines.org/sites/arrive/files/Author%20Checklist%20-%20Full.pdf (PDF). Please include your completed checklist as a Supporting Information file. Note that if your paper is accepted for publication, this checklist will be published as part of your article.

Reviewers' comments:

Reviewer's Responses to Questions

**Comments to the Author**

1. Is the manuscript technically sound, and do the data support the conclusions?

Reviewer #1: Yes

Reviewer #2: Yes

2. Has the statistical analysis been performed appropriately and rigorously? 

Reviewer #1: Yes

Reviewer #2: No

3. Have the authors made all data underlying the findings in their manuscript fully available?

Reviewer #1: Yes

Reviewer #2: Yes

4. Is the manuscript presented in an intelligible fashion and written in standard English?

Reviewer #1: Yes

Reviewer #2: Yes

5. Review Comments to the Author

Reviewer #1: The objective of the study is to assess the effect of Lignosus rhinocerotis Cooke Ryvarden on airway inflammation, mucus hypersecretion and airway hyper-responsiveness in a murine model of asthma. The manuscript is generally well written and the results are presented in good manner. The data supports the statements in results and discussion. However authors need to address the following.

1) The authors need to discuss the selection of the doses of LRE extract used in the present study.

2) Also need to provide the proper rationale for OVA induced asthma, preferably in the introduction.

3) More recent citations about inflammatory responses and airway hyper responsiveness to be added in the introduction.

Reviewer #2: This manuscript examined the effect of lignosus rhinocerotis(LR), a traditional medication, on an allergic asthmatic response in murine. The pretreatment of LR extract suppresses the airway inflammation and airway hyper responsiveness nicely as steroid. The results make significant contributions to applied asthmatic research.

1. This study is descriptive without any intervention design. The authors are encouraged to include more discussion points along possible mechanism.

2. The dose effect of LRE is required in the study.

3. What is the difference between those two allergic model? The rationale for the LRE applied to those similar model is not clear.

4. In ANOVA, the dependent variable must be a continuous. The statistic employed in the mucus score not appropriate.

5. What is the index for stability of the pharmaceutical product?

6. PLOS authors have the option to publish the peer review history of their article (what does this mean?). If published, this will include your full peer review and any attached files.

Reviewer #1: No

Reviewer #2: No

---

## [Author Response · Author response to Decision Letter 0]

19 Feb 2021

Journal Requirements:

 - Amended as required.

2. As part of your revision, please complete and submit a copy of the Full ARRIVE 2.0 Guidelines checklist, a document that aims to improve experimental reporting and reproducibility of animal studies for purposes of post-publication data analysis and reproducibility: https://arriveguidelines.org/sites/arrive/files/Author%20Checklist%20-%20Full.pdf (PDF). Please include your completed checklist as a Supporting Information file. Note that if your paper is accepted for publication, this checklist will be published as part of your article.

- Included on the submission as requested.

Reviewers' comments:

Reviewer #1: 

1) The authors need to discuss the selection of the doses of LRE extract used in the present study.

During allergen stimulation, dendritic cells induce the differentiation of T cells into Th2 cells. Th2 cells secrete cytokines i.e. IL-4, IL-5 and IL-13 which facilitate recruitment and activation of eosinophils in the airway. Meanwhile Th2 cells will induce the production of IgE by the B cells via IL-4 and IL-13 stimulation. Inflammatory mediators such as histamine and leukotrienes released by the eosinophils, T cells, macrophages, and neutrophils in the airway environment will result in damage to the airway, bronchoconstriction, and finally remodelling of the lung. In this study, oral administration of LRE at different doses (125, 250 and 500 mg/kg) showed varying responses. Among the doses tested in this study, LRE at 500 mg/kg significantly attenuated the level of IgE in serum, Th2 cytokines IL-4, IL-5 and IL-13 in BALF as well as leukocyte infiltrations in the lung tissues, in contrast to the other two lower doses, 125 and 20 mg/kg which were not consistently attenuated the responses. Based on the dose-response study, LRE 500 mg/kg was chosen for the subsequent studies.

2) Also need to provide the proper rationale for OVA induced asthma, preferably in the introduction.

Sensitisation methods utilising ovalbumin (OVA) and house dust mite (HDM) are known to enhance manifestations of asthmatic features. OVA- and HDM-induced model are considered the appropriate methods for experimental allergic asthma. These models have similar clinical symptoms to human asthma, which are characterized by airway mucosal oedema, bronchial wall thicknesses, mucus hypersecretion and increased infiltration of inflammatory cells into the lung. However, there has been limited success with OVA-induced model and only moderate pulmonary inflammation and mild AHR have been observed. HDM has become more commonly used in mouse models to induce AHR because it has immunogenic properties, so the use of an adjuvant is not required. In addition, inhaled delivery of HDM has been more successful in inducing AHR, possibly because of the intrinsic enzymatic activity of this allergen. 

3) More recent citations about inflammatory responses and airway hyper responsiveness to be added in the introduction.

During allergen stimulation, dendritic cells induce the differentiation of T cells into Th2 cells. Th2 cells secrete cytokines i.e. interleukin (IL)-4, IL-5, IL-9 and IL-13 which facilitate recruitment and activation of eosinophils in the airway. IL-5 and IL-9 are critical for promoting tissue eosinophilia and mast cell hyperplasia, whereas IL-13 stimulates mucus production by goblet cells and airway hyperresponsiveness (AHR). Meanwhile, Th2 cells will induce the production of IgE by the B cells via IL-4 stimulation. In the presence of airway inflammation, the cellular components especially the eosinophils and mast cells will be prompted to release a number of different mediators with the capacity to cause AHR. The inflammatory mediators such as IL-13, histamine, major basic proteins (MBP) and leukotrienes are known to cause AHR which lead to bronchoconstriction, and finally remodelling of the lung. 

Reviewer #2: 

1. This study is descriptive without any intervention design. The authors are encouraged to include more discussion points along possible mechanism.

Added as suggested. Please see in discussion section.

2. The dose effect of LRE is required in the study.

Some preliminary works to determine dose-response effect are included. 

Effects of LRE on IgE production in serum 

As shown in Fig. 1, the OVA-induced asthmatic rats (OVA group) presented significantly increased IgE levels compared to normal group. Significant decreased in the mean serum of IgE level was observed after treatments with LRE at 250 and 500 mg/kg (p < 0.05) when compared with OVA group. However, no significant decrease could be observed from the rats treated with LRE 125 mg/kg (Fig. 1). 

Effects of LRE on Th2 cytokines in BALF 

As shown in Fig. 2, the levels of IL-4, IL-5 and IL-13 were significantly increased in BALF of OVA group. Treatment with LRE 500 mg/kg resulted in significantly decreased of IL-4, IL-5 and IL-13 levels when compared to OVA group. In contrast, treatment with LRE 250 mg/kg significantly decreased only IL-5 level, while LRE 125 mg/kg significantly decreased IL-4 and IL-5 levels. 

Effects of LRE on histopathological change in lung tissue

Histopathological analysis by H&E staining demonstrated that lung tissue exposed with OVA had a marked increase in the leukocyte infiltration into the lung tissues. Treatment with LRE 250 or LRE 500 mg/kg significantly decreased infiltration of leukocytes in the peribronchial region and perivascular connective tissue of the airway compared with that observed in the OVA group (Fig. 3). No significant attenuation was observed by the LRE 125 mg/kg treatment group. Based on the dose-response study, LRE at the 500 mg/kg concentration was selected for the downstream experiments. 

3. What is the difference between those two allergic models? The rationale for the LRE applied to those similar model is not clear.

Some points on OVA and HDM models are included in the introduction. 

This study was started using ovalbumin-induced model. Using this model, we were able to induce a significant level of inflammation characterised by elevated IgE level, Th2 cytokines level, leukocytes infiltration and mucous in the lung tissue. However, when we tested AHR using OVA-induced model, we were not able to induce significant AHR level in the mouse model. Therefore, for AHR study, we used house dust mite (HDM) to induce the mice. The result as presented in this manuscript. 

4. In ANOVA, the dependent variable must be a continuous. The statistic employed in the mucus score not appropriate.

Data were expressed as mean ± standard deviation (SD). Statistical differences were assessed by one-way analysis of variance (ANOVA) test followed by Scheffe’s post hoc correction and Student’s t test using Statistical Programme for Social Science (SPSS) version 20.0 (New York, USA). p < 0.05 was considered statistically significant.

5. What is the index for stability of the pharmaceutical product?

The product shelf life is 5 years.

---

## [Decision Letter · Decision Letter 1]

5 Mar 2021

PONE-D-20-35276R1

Lignosus rhinocerotis Cooke Ryvarden ameliorates airway inflammation, mucus hypersecretion and airway hyper-responsiveness in a murine model of asthma

PLOS ONE

Dear Dr. Nurul,

Thank you for submitting your manuscript to PLOS ONE. After careful consideration, we feel that it has merit but does not fully meet PLOS ONE’s publication criteria as it currently stands. Therefore, we invite you to submit a revised version of the manuscript that addresses the points raised during the review process.

One reviewer still raised an issue regarding statistical method.

We look forward to receiving your revised manuscript.

Kind regards,

Yu Ru Kou, PhD

Academic Editor

PLOS ONE

Journal Requirements:

Reviewers' comments:

Reviewer's Responses to Questions

**Comments to the Author**

1. If the authors have adequately addressed your comments raised in a previous round of review and you feel that this manuscript is now acceptable for publication, you may indicate that here to bypass the “Comments to the Author” section, enter your conflict of interest statement in the “Confidential to Editor” section, and submit your "Accept" recommendation.

Reviewer #1: (No Response)

Reviewer #2: (No Response)

2. Is the manuscript technically sound, and do the data support the conclusions?

Reviewer #1: Yes

Reviewer #2: Yes

3. Has the statistical analysis been performed appropriately and rigorously? 

Reviewer #1: Yes

Reviewer #2: No

4. Have the authors made all data underlying the findings in their manuscript fully available?

Reviewer #1: Yes

Reviewer #2: Yes

5. Is the manuscript presented in an intelligible fashion and written in standard English?

Reviewer #1: Yes

Reviewer #2: Yes

6. Review Comments to the Author

Reviewer #1: (No Response)

Reviewer #2: The author’s responses to common # 5 do not reach my expectations. Please choose the right statistical test for mucus score which is discontinuous data.

7. PLOS authors have the option to publish the peer review history of their article (what does this mean?). If published, this will include your full peer review and any attached files.

Reviewer #1: No

Reviewer #2: No

---

## [Author Response · Author response to Decision Letter 1]

10 Mar 2021

Editor Comment:

We have uploaded all the figures in PACE and modified according to Plos ONE requirement. 

Reviewer Comment:

Thank you for the comment. For mucus score results, as the ordinal data are not normally distributed, Kruskal Wallis test is used as the appropriate test to compare the median among groups. The correct statistical test is mentioned and rephrased. 

Statistical analysis

 Data were expressed as mean ± standard deviation (SD). Statistical significance (p<0.05) was determined by a one-way ANOVA test followed by Scheffe’s post hoc correction using Statistical Programme for Social Science (SPSS) version 20.0 (New York, USA). For inflammation and mucus scoring, statistical analyses were performed using Kruskal Wallis tests with confidence interval adjustment by Bonferroni correction. The data from PCR array studies were analysed by the web-based software (SABiosciences) using a 2−∆∆CT method. The expression level for genes of interest was normalized by the expression of housekeeping genes.

---

## [Decision Letter · Decision Letter 2]

11 Mar 2021

Lignosus rhinocerotis Cooke Ryvarden ameliorates airway inflammation, mucus hypersecretion and airway hyper-responsiveness in a murine model of asthma

PONE-D-20-35276R2

Dear Dr. Nurul,

We’re pleased to inform you that your manuscript has been judged scientifically suitable for publication and will be formally accepted for publication once it meets all outstanding technical requirements.

Kind regards,

Yu Ru Kou, PhD

Academic Editor

PLOS ONE

Additional Editor Comments (optional):

Reviewers' comments:

Reviewer's Responses to Questions

**Comments to the Author**

1. If the authors have adequately addressed your comments raised in a previous round of review and you feel that this manuscript is now acceptable for publication, you may indicate that here to bypass the “Comments to the Author” section, enter your conflict of interest statement in the “Confidential to Editor” section, and submit your "Accept" recommendation.

Reviewer #2: All comments have been addressed

2. Is the manuscript technically sound, and do the data support the conclusions?

Reviewer #2: Yes

3. Has the statistical analysis been performed appropriately and rigorously? 

Reviewer #2: Yes

4. Have the authors made all data underlying the findings in their manuscript fully available?

Reviewer #2: Yes

5. Is the manuscript presented in an intelligible fashion and written in standard English?

Reviewer #2: Yes

6. Review Comments to the Author

Reviewer #2: (No Response)

7. PLOS authors have the option to publish the peer review history of their article (what does this mean?). If published, this will include your full peer review and any attached files.

Reviewer #2: No

---

## [Editor Report · Acceptance letter]

19 Mar 2021

PONE-D-20-35276R2 

*Lignosus rhinocerotis* Cooke Ryvarden ameliorates airway inflammation, mucus hypersecretion and airway hyperresponsiveness in a murine model of asthma 

Dear Dr. Nurul:

I'm pleased to inform you that your manuscript has been deemed suitable for publication in PLOS ONE. Congratulations! Your manuscript is now with our production department. 

Kind regards, 

on behalf of

Dr. Yu Ru Kou 

Academic Editor

PLOS ONE